# Epigenetic Marks, DNA Damage Markers, or Both? The Impact of Desiccation and Accelerated Aging on Nucleobase Modifications in Plant Genomic DNA

**DOI:** 10.3390/cells11111748

**Published:** 2022-05-25

**Authors:** Beata P. Plitta-Michalak, Monika Litkowiec, Marcin Michalak

**Affiliations:** 1Department of Chemistry, University of Warmia and Mazury in Olsztyn, Plac Łódzki 4, 10-719 Olsztyn, Poland; 2Kostrzyca Forest Gene Bank, Miłków 300, 58-535 Miłków, Poland; monika.litkowiec@gmail.com; 3Department of Plant Physiology, Genetics and Biotechnology, University of Warmia and Mazury in Olsztyn, Oczapowskiego 1A/103, 10-719 Olsztyn, Poland; m.michalak@uwm.edu.pl

**Keywords:** *Acer pseudoplatanus*, biomarker, desiccation, DNA oxidative damage, ELISA, embryonic axes, hm^5^C, m^5^C, 8-oxoG, recalcitrant seeds, sycamore, viability

## Abstract

Modifications of DNA nucleobases are present in all forms of life. The purpose of these modifications in eukaryotic cells, however, is not always clear. Although the role of 5-methylcytosine (m^5^C) in epigenetic regulation and the maintenance of stability in plant genomes is becoming better understood, knowledge pertaining to the origin and function of oxidized nucleobases is still scarce. The formation of 5-hydroxymetylcytosine (hm^5^C) in plant genomes is especially debatable. DNA modifications, functioning as regulatory factors or serving as DNA injury markers, may have an effect on DNA structure and the interaction of genomic DNA with proteins. Thus, these modifications can influence plant development and adaptation to environmental stress. Here, for the first time, the changes in DNA global levels of m^5^C, hm^5^C, and 8-oxo-7,8-dihydroguanine (8-oxoG) measured by ELISA have been documented in recalcitrant embryonic axes subjected to desiccation and accelerated aging. We demonstrated that tissue desiccation induces a similar trend in changes in the global level of hm^5^C and 8-oxoG, which may suggest that they both originate from the activity of reactive oxygen species (ROS). Our study supports the premise that m^5^C can serve as a marker of plant tissue viability whereas oxidized nucleobases, although indicating a cellular redox state, cannot.

## 1. Introduction

DNA is a highly dynamic molecule that is continuously subjected to events that may result in alterations in the nucleobases and the DNA backbone structure. The four canonical bases that make up genomic DNA, adenine (A), thymine (T), guanine (G), and cytosine (C), are subjected to a wide variety of chemical modifications. While some of these alterations reflect regulated modifications of DNA, others represent DNA damage resulting from superimposed events, such as aging. Research conducted over the past decade has resulted in the identification of the enzymes and non-enzymatic agents responsible for induction and processing of DNA modifications [1,2].

One of the well-known DNA modifications is the methylation of cytosine which was initially observed and reported more than a century ago [3]. The presence of 5-methylcytosine (m^5^C) in DNA was demonstrated by Hotchkiss [4] in 1948, and seven decades of subsequent studies have clearly established that the nucleotide sequence is not the only information contained in DNA. Notably, it is the most common DNA modification in eukaryotes and often referred to as the “fifth base”. The m^5^C modification represents a conserved epigenetic mark that plays a role in many essential biological processes, including heterochromatin formation, cell proliferation, genomic imprinting, regulation of endogenous gene expression, as well as defense against transposon and silencing transgenes [5]. The total level of cytosine methylation of DNA ranges between 5–25%, depending on the plant species [6,7,8,9,10,11,12].

DNA methylation is catalyzed by DNA methyltransferases, which utilize the methyl group from S-adenosyl-L-methionine. In plants, m^5^C is found in the context of CG, CHG, and CHH sequences (where H represents A, C, or T). The de novo induction and maintenance of methylation are associated with different pathways. The first, which occurs in all sequence contexts, is driven by RNA (RNA-directed DNA methylation, RdDM) and catalyzed by DOMAINS REARRANGED METHYLASE 2 (DRM2). Post-replication maintenance of the methylation event within CHH and CHG sequences is regulated by DRM2, CMT2 (CHROMOMETHYLASE 2), and CMT3 (CHROMOMETHYLASE 3). Lastly, CG dinucleotides are methylated by MET1 (METHYLOTRASNFERASE 1), which recognizes hemimethylated DNA [13]. 

Notably, 5-methylcytosine can be passively removed from DNA when DNA methyltransferase activity is absent, or when the source for the methyl group donor is deficient. Active demethylation occurs by base excision (base excision repair, BER), which is carried out by enzymes involved in DNA repair. In contrast to the single enzyme involved in m^5^C synthesis, the removal of m^5^C requires the activity of several enzymes and is initiated by DNA demethylases. While m^5^C removal in mammals is accomplished by DNA demethylation involving m^5^C oxidation and/or deamination, plants can directly remove m^5^C from any sequence context [13]. Arabidopsis thaliana has four active m^5^C DNA demethylases: ROS1 (REPRESSOR OF SILENCING 1), DME (TRANSCRIPTIONAL ACTIVATOR DEMETER), DML2, and 3 (DEMETER-LIKE PROTEIN) [6]. Currently, DNA methylation is one of the best-characterized epigenetic processes, and m^5^C patterning when properly established and maintained, is essential for the normal functioning of living cells [7].

In addition to the normal, regulated modifications of DNA, like m^5^C, that are responsible for epigenetic control, DNA modifications can also be induced by environmental and endogenous factors resulting in DNA damage. In particular, reactive oxygen species (ROS), among the multiple agents that damage DNA, deserve special attention due to their omnipresence and their ability to impact DNA structure and function [1,14,15,16,17,18]. ROS have a high potential to cause strand breaks (SB) in DNA that severely impact genomic integrity [19]. In most cases, superoxide anion radicals (O_2_^−^) are the primary type of ROS produced and are readily converted to H_2_O_2_ and other peroxides. H_2_O_2_ is the most stable type of ROS and can freely migrate across cell membranes. Peroxides are involved in the metal-catalyzed Haber–Weiss reaction in the presence of reduced transition metals (Fe^2+^; Fenton reaction) [20] and Cu^+^, which yield hydroxyl radicals (OH), the strongest type of cellular oxidant [1,14,21,22,23]. Additionally, metal-independent ^·^OH formation has also been established, which occurs by a mechanism that relies on polyhalogenated quinone-mediated H_2_O_2_ homolytic decomposition [24]. The OH radical reacts immediately with every molecular species in the vicinity and its activity is only restricted by its diffusion rate. This radical can form adducts with all four bases, and in the process, extracts a hydrogen atom from 2′-deoxyribose, leading to strand scission, base release, and/or oxidation. Notably, guanine (G) is the most sensitive of the nucleobases to oxidation as it has the lowest redox potential. As a result, 8-oxo-7,8-dihydroguanine (8-oxoG) represents the most abundant oxidative DNA base lesion, and can arise through the activity of ^·^OH, hydroperoxide radicals (OOH), or rare singlet oxygen (^1^O_2_) [9,10,16,25]. Peroxynitrite anions (ONOO^−^), a reactive nitrogen species (RNS), can also form 8-oxoG, although at a slower rate than by other methods due to an unfavorable reaction barrier [25]. The most recent reports indicate, however, that carbonate radicals (CO_3_^−^) rather than ^·^OH radicals are the major product of the Fenton reaction, and that the carbonate radical specifically generates C8 or C5-oxidized G but does not cause direct DNA strand breaks [16,21,22,26,27,28]. These new findings deserve further study, as they generate new questions and ideas about the plausible regulatory function of 8-oxoG. Despite the fact that 8-oxoG is efficiently removed from DNA by BER and NER (nucleotide excision repair), it can still be detected and used as an indicator of DNA exposure to ROS [18,29].

The compound 5-hydroxymethylcytosine (hm^5^C) is a new epigenetic mark that has received tremendous attention in epigenetic research [5]. It was first identified in bacteriophages by Wyatt and Cohen (1952) [30], and in vertebrates in 1972 [31], but its function was poorly understood until 2009 when it was shown to comprise 0.6 and 0.2% of the nucleotides in Purkinje and granule cells [32,33]. Those studies revitalized the interest in hm^5^C and it is now often referred to as the “sixth base” [5,6]. It is currently believed that this DNA modification plays a critical role in many biological processes in eukaryotes [5,6].

While active DNA demethylation in plants can be achieved through direct cleavage of DNA by DNA glycosylases, followed by the replacement of m^5^C with C utilizing BER machinery, studies in mammals have demonstrated that m^5^C can be sequentially oxidized to hm^5^C, 5-formylcytosine (fo^5^C), and 5-carboxycytosine (ca^5^C) by members of the TET/JBP (ten-eleven translocation/J-binding proteins) family of iron-(II)/α-ketoglutarate-dependent dioxygenases [7,34,35]. The m^5^C oxidation products are subsequently recognized and cleaved by thymine-DNA glycosylase (TDG), thus, restoring unmethylated C via the BER pathway [7]. The consecutive oxidation of m^5^C in mammalian genomes raises a question pertaining to the potential presence of consecutive oxidation products in plant genomes [8]. In this regard, hm^5^C has been detected and reported in several plant genomes, with levels estimated to range between 0.068 and 0.075% in Arabidopsis, soybean, and rice DNA [36,37,38,39,40]. The origin of this modification is still in question, however, because TET/JBP proteins and UHRF2 (ubiquitin-like PHD and Ring Finger Domains 2), the proteins associated with hm^5^C, have not been identified in plants [6,34,38,41]. Therefore, it is assumed that hm^5^C may not be a product of enzymatic activity. In contrast, Tang et al. [7] demonstrated the presence of fo^5^C and ca^5^C in genomic DNA of various plant tissues and suggested that in addition to direct DNA cleavage by glycosylase, active DNA demethylation in plants may be achieved through an alternative pathway similar to mammals. Finally, the formation of hm^5^C and fo^5^C has been proposed to arise from H-atom extraction from the methyl group of m^5^C by ^·^OH [42]. As a result, a clear distinction between hm^5^C as a plant epigenetic mark or marker of DNA damage is still unresolved.

Therefore, in order to address this issue in the present study, we examined the changes in m^5^C, hm^5^C, and 8-oxoG in embryonic axes of sycamore maple (*Acer pseudoplatanus* L.) under desiccation stress conditions and after accelerated aging. We previously used sycamore seeds as a model in research on recalcitrant (desiccation-sensitive) seeds [10,43]. Here, we report for the first time on the changes in the level of three DNA modifications (m^5^C, hm^5^C, and 8-oxoG) while simultaneously tracking the level of ROS and antioxidants (AOX). We discuss the nature of those modifications and the implications of their-genome wide imbalances, as well as why only global methylation status is a good biomarker of tissue viability.

## 2. Materials and Methods

### 2.1. Plant Material and Treatments

Samaras from sycamore (*Acer pseudoplatanus* L.) were collected from trees in provenances located in Olsztyn (53.7553° N, 20.4561° E), in northeastern Poland. The collected samaras were dried at room temperature and 35% RH (relative humidity) and stored at 3 °C in tightly sealed, plastic containers. Embryonic axes were excised from the seeds no later than eight weeks after collection. The desiccation of embryonic axes was conducted as previously described [9,10,43]. The duration of the desiccation protocol ranged from 1 to 18 h. and was calculated on a fresh weight basis as previously described [44] (Table 1). In the accelerated aging experiment (storage at high temperature and high MC), fresh embryonic axes at a MC of 50% were kept in 2 mL tubes at 45 °C for three days. Unless otherwise indicated, all chemicals and materials were purchased from Merck (Darmstadt, Germany).

### 2.2. Viability Assessment

A tetrazolium chloride (TTC) assay was used to assess the viability of axes subjected to accelerated ageing as previously described [43,44,45]. Four biological replicates comprising ten embryonic axes in each replicate were stained in 1% 2,3,5-triphenyltetrazolium chloride (TTC) solution. Embryonic axes were considered dead when remaining green, while those stained pink to red were classified as metabolically active and alive.

An in vitro regrowth assay was conducted as previously described [43] using five biological replicates comprising 10–15 embryonic axes in each replicate. After eight weeks, regrowth of embryonic axes was assessed and only counted as regenerated if the embryonic axis had developed either a shoot or a shoot with a root.

### 2.3. ROS Detection

ROS levels were quantified in aged embryonic axes using the fluorogenic dye, 2′,7′- dichlorodihydrofluorescein diacetate (H_2_DCFDA; Invitrogen, Waltham, MA, USA) as previously described [43]. Embryonic axes were incubated in 1 mL of 10 µM H_2_DCFDA in darkness for 15 min. Fluorescence was measured at an excitation wavelength of 492 nm and an emission wavelength of 525 nm using an Infinite M200 PRO plate reader (Tecan, Männedorf, Switzerland). Four replicates comprising five embryonic axes in each replicate were analyzed. Results are expressed as relative fluorescence units per gram of dry weight (RFU g^−1^ DW).

### 2.4. Measurement of Total and Non-Protein Antioxidant Capacity

Total antioxidant capacity (TAC) was measured using a commercially available Total Antioxidant Capacity Assay kit (Merck (Darmstadt, Germany). Five embryonic axes were used as replicates in each experiment and each experiment was replicated four times. Explants were homogenized in liquid nitrogen (LN) and samples were extracted in 0.8 mL of ice cold 1X Phosphate Buffered Saline (PBS) to maintain the recorded antioxidant values within the range of the standards provided in the commercial kit. Absorbance of the samples (10 µL) was assessed at λ 570 nm according to the manufacturer’s instructions. Antioxidant capacity was calculated based on trolox equivalent antioxidant capacity, that is, the amount of trolox producing the same effect as the sample studied. Calculations were made based on standard curves obtained for a trolox solution. Values were then normalized to tissue dry weight. For non-protein antioxidant capacity (NPAC), samples were diluted 1:1 with Protein Mask provided by the manufacturer.

### 2.5. DNA Isolation and Measurement of DNA Modifications

Total genomic DNA was extracted from control embryonic axes, embryos subjected to desiccation, and embryos subjected to accelerated ageing after the axes were homogenized in LN using a NucleoSpin Plant II Kit (Macherey-Nagel, Düren, Germany) according to the manufacturer’s instructions. Five embryonic axes were used as replicates in each experiment and each experiment was replicated four times. DNA concentration and quality (A_260_/A_280_ = 1.8–1.9) were measured with a NanoQuant Plate M200 PRO (Tecan, Männedorf, Switzerland). DNA modifications were quantified colorimetrically using MethylFlash Global DNA Methylation (5-mC), Hydroxymethylation (5-hmC) ELISA Easy Kits, and EpiQuik 8-OHdG DNA Damage Quantification Direct Kit (Epigentec, Farmingdale, NY, USA) according to the manufacturer’s instructions. Two technical replicates of each biological replicate were measured. A total of 100 ng of DNA was used for the m^5^C and hm^5^C measurements, while 200 ng was used for the 8-oxoG measurements. The relative quantity of the different entities was calculated according to the formula provided in the commercial kits using the provided standards for the preparation of standard curves.

### 2.6. Statistical Analysis

Graphical visualization and statistical analysis of the obtained data was conducted with R software (R Core Team 2020; https://www.r-project.org; accessed on 1 April 2022). The effect of desiccation on viability (TTC staining assay, regrowth) was evaluated separately using a general linear model (GLM) with a binomial distribution. The impact of desiccation and accelerated aging on the level of m^5^C, hm^5^C, and the assessment of the total amount of antioxidants, were evaluated using a linear model. An assessment of normality of the data was conducted using the Shapiro–Wilk test, and the Levene’s test was used to test homogeneity of variance. For non-normally distributed data or data with non-homogenous variation, such as the level of 8-oxoG, ROS, and the amount of non-protein antioxidants, a non-parametric Kruskal–Wallis test was used. A one-way ANOVA was used to test significant differences between mean values of data that were normally distributed. Pairwise comparisons between treatments were performed using a Duncan’s multiple range test at *p* ≤ 0.05. The correlation between in vitro regrowth and viability measured by TTC, and the levels of hm^5^C, m^5^C, 8-oxoG, ROS, the assessment of the total amount of antioxidants and the amount of non-protein antioxidants after different desiccation time and accelerated aging, were analyzed using a Spearman correlation coefficient analysis. An R-package, ‘corrplot‘, was used for the analysis and construction of the correlation between the measured parameters. Principal component analysis (PCA) was applied to z-scores of the data. R-packages ‘ggplot2’, ‘factoextra’ and ‘FactoMineR’, were used for the principal component analysis (PCA), while ‘ggplot2’ was used for the graphical visualization of the data.

## 3. Results

### 3.1. Explant Viability: TTC Assay and Regrowth

The impact of desiccation on the viability of embryonic axes of *A. pseudoplatanus* was assessed by the level of metabolic activity of the tissues determined by TTC staining assay and as regrowth of seedlings after 8 weeks of in vitro culture. Data obtained from a previous study [43], along with data for metabolic activity of embryonic axes subjected to accelerated ageing, are also presented (Figure 1). Altogether results indicated that a desiccation period of four and six hours significantly decreased the viability of embryonic axes while explants desiccated for 18 h, or subjected to accelerated aging, were dead.

### 3.2. Measurement of Reactive Oxygen Species

The impact of gradual desiccation and accelerated aging on ROS was determined using the H_2_DCFDA assay. Previously presented data, along with data for embryonic axes subjected to accelerated aging, are also presented. The highest level of ROS was detected in explants subjected to accelerated aging for 3 days at 45 °C (1440 RFU g^−1^ DW), (Figure 2).

### 3.3. Measurement of Total Antioxidant Capacity and Non-Protein Antioxidant Capacity

The highest total antioxidant capacity (TAC) (38.4 nmol/µL g^−1^ DW) was observed in control embryonic axes. Desiccation of embryonic axes for 1 h resulted in only a slight decrease in the level of TAC (36.1 nmol/µL g^−1^ DW). Further desiccation for 4 h, 6 h, and 18 h caused a significant progressive decline in TAC to 34.4, 31.9, and 26.7 nmol/µL g^−1^ DW, respectively (Figure 3). The lowest level of TAC (25.8 nmol/µL g^−1^ DW) was detected in embryonic axes subjected to accelerated aging.

Non-treated (control) embryonic axes of *A. pseudoplatanus* exhibited the highest non-protein antioxidant capacity (NPAC) (26.7 nmol/µL g^−1^ DW). A similar level was observed after 1 h desiccation, however, after desiccation for 4 h and 6 h, a decrease in NPAC was detected (24.4 and 24.2 nmol/µL g^−1^ DW, respectively) (Figure 3). The NPAC after desiccation for 18 h and accelerated ageing was comparable (19.6 nmol/µL g^−1^ DW and 18.7 6 nmol/µL g^−1^ DW, respectively).

### 3.4. Assessment of the Genomic Level of m^5^C, 8-oxoG, and hm^5^C in the DNA of Embryonic Axes of Acer pseudoplatanus

Non-treated (control) embryonic axes of *A. pseudoplatanus* exhibited the highest level of m^5^C (11.9%) (Figure 4A). As desiccation time increased, a significant decrease in the level of m^5^C in DNA was observed to 10.2%, 9.48%, 8.11%, and 6.96% after desiccation for 1, 4, 6, 18 h, respectively. Accelerated aging had the strongest impact on the level of global m^5^C, reducing it to half the level (5.56%) that it was in control embryonic axes.

The lowest percentage of 8-oxoG in DNA (0.0053%) was detected in non-treated, control embryonic axes. Desiccation of embryonic axes for 1 h caused a significant increase in the detected amount of 8-oxoG, increasing the percentage to 0.022% (Figure 4B). Further desiccation resulted in a decline in 8-oxoG to 0.0112%. No difference was observed between the percentage of 8-oxoG in embryonic axes desiccated for 6 and 18 h (0.0137% and 0.0135%, respectively). In contrast, a significant 4.6-fold increase in the level of 8-oxoG, relative to the control, was observed in embryonic axes after 3 days of accelerated aging at 45 °C.

A global hm^5^C level of 0.102% was detected in control embryonic axes (Figure 4C). Desiccation for 1 h induced a significant increase in the detected level of hm^5^C to 0.128%. After desiccation of embryonic axes for 4 h, however, a significant decrease in the level of this modification to 0.041% was observed. Notably, further desiccation for 6 h and 18 h, as well as accelerated ageing conditions, did not significantly affect the level of hm^5^C, which remained in the range of 0.075–0.081%.

### 3.5. Correlation and Principal Component Analyses

Spearman correlation coefficient analysis of the detected levels of m^5^C, hm^5^C, 8-oxoG, ROS, TAC, and NPAC indicated that the level of 8-oxoG was significantly negatively correlated with the level of m^5^C, TAC, and NPAC (Figure 5). The level of 8-oxoG was also significantly positively correlated with the detected level of ROS. The level of m^5^C exhibited a highly significant negative correlation with the level of ROS, and a highly significant strong positive correlation with TAC and NPAC. No significant correlation was observed between the level of hm^5^C and any of the other measured parameters.

The PCA results (Figure 6) revealed a negative relationship between the viability of embryonic axes (as measured by either metabolic activity or regrowth), the level of m^5^C, TAC, and NPAC, with the level of 8-oxoG and ROS. This negative relationship was reflected in the opposite coordinates of these two groups and the high loading of these variables in principal component 1 (Prin 1), (Appendix A). The level of m^5^C, TAC, NPAC, metabolic activity of the tissues (TTC), and regrowth of seedlings in in vitro culture (RG) had the biggest influence on principal component 1 (Prin 1). In contrast, hm^5^C and 8-oxoG had the biggest influence on principal component 2. This influence was reflected in the high presence of these variables in Prin1 and Prin2, respectively (Appendix A). The first two principal components accounted for 66.9 and 19.5% of the observed variance, respectively. The embryonic axes clustered into six groups based on Prin1 and Prin2, clustering in groups corresponding to their treatment. The results indicated that the groups differed mostly based on their level of m^5^C, TAC, and NPAC.

## 4. Discussion

Seeds of *Acer pseudoplatanus* L. (sycamore) are desiccation sensitive, a feature that complicates their long-term storage since the seeds do not tolerate desiccation below 30% of MC. Owing to their recalcitrance, sycamore seeds can only be stored under standard conditions for 2–3 years at −3 °C. Thus, sycamore seeds have become a model system for desiccation studies of temperate zone recalcitrant seeds [10,43,46,47,48,49,50]. Notably, the predominant categorization system utilized to classify seeds is based on their tolerance to water withdrawal [51]. Desiccation studies have received considerable attention by seed researchers since desiccation tolerance is a determining factor for successful seed preservation under both conventional and cryogenic conditions [12,52,53,54].

The current study was conducted to provide information on the changes in nucleobase modifications that occur in seeds when they are exposed to the stress of desiccation. The objective was to determine the changes that occur in the global level of three nucleobase modifications in embryonic axes in response to desiccation, to better understand the relationship between these modified nucleobases. Notably, we were able to document for the first time changes in the global level of hm^5^C under conditions that have not been previously examined. To determine if the observed changes are characteristic only for tissues with decreasing viability due to desiccation, we also measured changes in these three nucleotide modifications in embryonic axes subjected to accelerated aging at high moisture content and temperature. This methodology was used to enable us to simulate, over a short period of time, natural aging processes that occur in plant tissues over an extended time period.

We previously used 2D-TLC to monitor changes in global DNA methylation in seeds, which allowed us to measure the level of m^5^C in DNA [10,11,44], even in problematic seed samples that are rich in lipids and proteins. Although this method has several notable advantages, it does not allow for quick, high-throughput analyses, preferably performed in a 96-well format, that provides the ability to assess multiple biological samples simultaneously in a statistically sound manner. Therefore, we decided to use an enzyme-linked immunosorbent assay (ELISA) in the present study. This method has been successfully used by others [55,56,57], although it has never been used in the context of recalcitrant seed desiccation studies.

Desiccation induces oxidative stress and a redox imbalance. Cells, however, possess a system of enzymatic proteins (dismutase, peroxidases, catalase), non-enzymatic proteins (peroxiredoxins, thioredoxins, glutaredoxins, and metallothionein), and low molecular weight antioxidants (glutathione ascorbate, carotenoids) [1] to prevent the formation and accumulation of ROS and to maintain the redox equilibrium. In the present study, the total, as well as non-protein small molecule antioxidant capacity, decreased with time of desiccation and in samples subjected to accelerated aging. We tracked ROS levels in a previous study using the fluorescent probe H_2_DCFDA, which is oxidized by multiple oxidative agents [43], thus, providing information on the oxidative stress level present in plant tissues. In the present study, we measured both the oxidative and antioxidant status of cells at each stage of desiccation and after accelerated aging, along with concomitant changes in the level and type of DNA modifications. As a result, we demonstrated a correlation between the redox state of tissues and the 8-oxoG and m^5^C level, although no clear correlation was observed for hm^5^C. Further we explain, nevertheless, why we consider hm^5^C as a result of oxidation.

Chemical modifications to DNA add an additional layer of complexity to cellular processes due to their ability to function in a regulatory network that modulates chromatin structure and genome function. Cytosine methylation is an element of the epigenome, a system of non-sequence based, potentially heritable traits. The composition of the epigenome within a given cell is a function of both genetic determinants and external factors [6,7,58]. In this regard, desiccation stress has been reported to induce changes in the pattern and level of DNA methylation in various plant tissues, including recalcitrant and orthodox (desiccation resistant) seeds [9,10,44]. Our current results confirm previous observations. Rapid, severe desiccation of isolated recalcitrant embryonic axes resulted in a decrease in the percentage of total m^5^C in genomic DNA. This decline was highly correlated with a decline in viability (Figure 5). Notably, the reduction in m^5^C was induced by both desiccation and accelerated aging, therefore treatment differing in tissue moisture content and temperature. We previously demonstrated that the decrease in viability of *Quercus robur* L. recalcitrant seeds during storage (natural aging) also resulted in a decline in global m^5^C levels [11]. Additionally, it was also reported that epigenetic changes occurred upon accelerated aging conditions in orthodox seeds of *Secale cereale* L. and *Mentha aquatica* L., even before a major loss of viability was observed [59,60]. Therefore, collectively the data indicate that the decline in DNA methylation that occurs in plant embryos that also undergo a reduction in viability is a universal process resulting from being subjected to different stresses (water withdrawal vs. high temperature and high MC) [11,61]. As a result, reductions in m^5^C levels can be considered as a viability marker, which was proved by high significant correlation (Figure 5) and PCA analysis (Figure 6).

Pro-oxidant species are relatively abundant within cells and are continuously generated by endogenous metabolism and in response to exposure to external stress factors [29]. ROS-mediated DNA oxidation represents a threat to the stability of genetic material and has been estimated to occur at a rate that is only 50% lower than that occurring for protein oxidation [29,62]. While ROS can oxidize DNA either by altering the four bases or deoxyribose, the most stable and frequent DNA oxidation product in DNA is 8-oxoG [29], and as a result, this alteration is considered as a marker for oxidative damage to DNA. In the present study, the changes in 8-oxoG were the result of oxidative stress induced by desiccation and accelerated aging. We previously analyzed desiccation-driven DNA damage and repair after rehydration of seeds using an enzyme-modified alkaline comet assay with formamidopyrimidine [fapy]-DNA glycosylase (Fpg) enzyme, which recognizes oxidized forms of guanine [44]. By using another method in the present study, we confirmed that elevated levels of oxidative stress induced by desiccation negatively impact DNA integrity. A high percentage of oxidized G was also detected in embryonic axes that lost their viability due to accelerated ageing. Notably, 8-oxoG was also reported to be the most common and abundant DNA lesion in mammalian aged cells [63]. Accumulation of ROS resulting in 8-oxoG increase in DNA and lower seed viability was also reported [64,65,66]. Moreover, transcriptional changes during seed ageing were accompanied by a progressive shift towards a more oxidizing cellular environment and nucleic acid degradation, which occurred prior to any loss of seed viability [67]. However, in current research, in contrast to the m^5^C epigenetic mark, high levels of oxidative damage can clearly be detected in viable tissue affected by desiccation-driven oxidative stress, as well as in dead tissue. Therefore, an increase in 8-oxoG does not appear to be specific to the physiological state of the tissue, which suggests that one needs to track DNA repair efficiency over time to distinguish viable and non-viable tissues [43]. Therefore, the evaluation of 8-oxoG as a single time point measurement performed immediately after any treatment may not represent an accurate assessment of viability. This premise agrees with previous research [68] showing that biomarkers such as ROS are useful in describing the nature of stress and the reason for cellular damage but may not be helpful as direct markers of the actual viability state of a tissue.

Nevertheless, oxidative damage to G presents a severe threat to genome and epigenome stability. It represents a pre-mutagenic lesion as its *syn* arrangement pairs with adenine leading to a G:C-T:A transversion upon replication [29,69]. In vitro experiments have demonstrated that the presence of 8-oxoG destabilizes DNA fragments possessing different methylation patterns. The dynamic and thermodynamic effects are not additive in fully methylated oxidized CpG, indicating that the introduced modifications interact with each other. The close proximity of m^5^C and 8-oxoG within a major groove, together with steric and ion-dipole interactions between 8-oxoG and the sugar-phosphate moiety of the 8-oxoG nucleotide, may be expected to disturb the conformational dynamics of methylated and oxidized CpG sites [69]. Finally, 8-oxoG is known to interfere with the action of many DNA-binding proteins. Thus, the conformational and dynamic effects of spurious DNA oxidation in the regulatory CpG dinucleotides can have a far-reaching biological impact [69]. The presence of 8-oxoG was shown to bias the recognition of methylated CpG dinucleotides by ROS1, with an approximately fourfold preference for the oxidized DNA strand [69]. Additionally, 8-oxoG was also shown to affect cytosine methylation by interfering with the binding of DNA methyltransferases [29]. Consequently, a negative correlation between m^5^C content and 8-oxoG was observed and was also evident in our present study (Figure 5). Indeed, cells that are deficient in the antioxidant enzyme CuZn-SOD exhibit an accumulation of 8-oxoG and massive levels of DNA hypomethylation [29,70,71]. Therefore, it seems that oxidative DNA damage has a substantial impact on epigenetic markers by blocking their synthesis and enhancing their removal.

Interestingly, it was recently suggested that 8-oxoG can play a role in the regulation of genomic functions through mechanisms responsible for its formation and processing [29]. Epigenetics studies focus on heritable changes in genome function that are not due to alterations in the DNA sequences. In this regard, m^5^C is a common epigenetic mark that provides a new level of epigenetic information, while 8-oxoG also represents a stable change in DNA structure similar to m^5^C and may provide an additional regulatory influence on the primary sequence that is dictated by the redox environment surrounding the genome [29]. In this regard, 8-oxoG is produced by fluxes of H_2_O_2_ from the cytosol to the nucleus, thus reflecting the intracellular redox balance. It can also be produced, however, by a local reaction involving the formation of ^·^OH from H_2_O_2_ by iron-containing complexes associated with DNA structure in close proximity to chromatin. H_2_O_2_ is also sourced through the activity of nuclear oxidases and FAD-dependent lysine-specific histone demethylases (LDL1, LDL2, LDL3 FLD) [1,72] that participate in the regulation of gene expression [1,29]. It is still not known if these two pathways (cytosolic and nucleolar) cooperate and if 8-oxoG encodes transmissible information capable to durably alter cellular behavior [29].

The last modification we tracked was hm^5^C, which is also recognized as a stable epigenetic mark [35]. In general, oxidized forms of m^5^C in plants are very limited and there is no consensus currently if the oxidation is a result of ROS or enzymatic activity [5,7,8,34,41]. The relatively low levels of m^5^C oxidation products suggest that they most likely arise due to the activity of ROS in cells, which is supported by the lack of any functional Tet-family dioxygenase enzymes (writer and primary eraser of hm^5^C) and UHFR2 (reader of hm^5^C) [5,8,35] being identified. Moricova et al., 2013 [55], however, detected hm^5^C in protoplasts of *Cucumis sativus* L. and *Brassica oleracea* L. but stated that oxidative damage was not the main source of hm^5^C. Only a few studies have investigated the role of hm^5^C in plants under stress conditions [73]. In the present study, the changes in global hm^5^C induced in the genome of *A. pseudoplatanus* by progressive steps of desiccation mirrored the changes observed for guanine oxidative damage (Figure 4). An initial increase in hm^5^C was observed after 1 h of desiccation when the change in MC was the highest, followed by a drop after 4 h, in association with corresponding DNA repair in viable tissues [43], followed by a plateau in tissues with low viability or that were dead (after 6 and 18 h of desiccation, respectively). Although the initial hm^5^C increase is smaller in comparison to 8-oxoG, the similarity between these changes in global DNA modification during subsequent desiccation steps is striking. However, when the data obtained from embryonic axes subjected to accelerated ageing were analyzed, concomitant increases in ROS and 8-oxoG were clearly evident, while no such increase in hm^5^C was observed in embryonic axes subjected to accelerated aging. This is contrary to what one would expect for oxidative DNA damage resulting from increased levels of ROS (Figure 2), however it is in concordance with previous studies, as it has been reported that there is no correlation between the age of adult mice and the level of hm^5^C in Purkinje and granule cells [32], while a decrease in hm^5^C was recently reported in mouse cells and tissues subjected to physiological and accelerate aging [74]. Therefore, it could be hypothesized that the level of hm^5^C can be affected by the accumulation of its oxidized forms (fo^5^C and ca^5^C), which have been detected in plant genomes [7]. Environmental stresses, such as drought and salinity, can alter the level of fo^5^C and ca^5^C in plant genomes, suggesting that these modifications play a role in plant response to environmental stress [7]. A second potential scenario is that a decrease of m^5^C, resulting from demethylation or 8-oxoG proximity (discussed above), may also reduce the amount of hm^5^C since it is derived from m^5^C. Indeed, a decrease in m^5^C was observed in aging plant tissues concomitant with a decrease in CMT3 and MET1 expression, and increased expression of ROS1 [75]. This suggests that regions of DNA in plants may undergo demethylation during plant aging due to a decrease in DNA methylation and the activation of DNA demethylation processes [75,76]. Therefore, the lack of correlation between ROS and hm^5^C (Figure 5) may result from aging-related m^5^C decline. Moreover, hm^5^C does not appear to be a direct marker of tissue viability (Figure 5), although it may exert an impact on genome stability and gene expression. For example, hm^5^C in plants may be involved in passive demethylation during cell division. The *A. thaliana* VIM1 protein has been reported to recognize hm^5^C in vitro, and it has been suggested that hm^5^C at a CpG site may trigger VIM-mediated passive loss of cytosine methylation during *Arabidopsis* DNA replication [36]. In addition, plants possess a DNA glycosylase that can directly cleave m^5^C, and both DME and ROS1 can also react with hm^5^C [77], also participating in the demethylation process by removal of oxidized m^5^C.

## 5. Conclusions

Our data revealed and confirmed the presence of modified DNA bases in the sycamore genome. Our study supports previous reports that desiccation induces oxidative stress and leads to a decline in the pool of antioxidants and increase in ROS. For the first time, however, we demonstrated that these changes occur along with both a decrease in global DNA methylation and concomitant mirroring changes in the level of oxidized m^5^C and G in embryonic axes of sycamore subjected to desiccation stress, suggesting that generated ROS react with both G and m^5^C and that hm^5^C may serve as both an epigenetic mark and an oxidative stress marker. However, only m^5^C can be considered as a viability marker, as we did not detect any significant correlation between 8-oxoG or hm^5^C and tissue viability.

## Figures and Tables

**Figure 1 cells-11-01748-f001:**
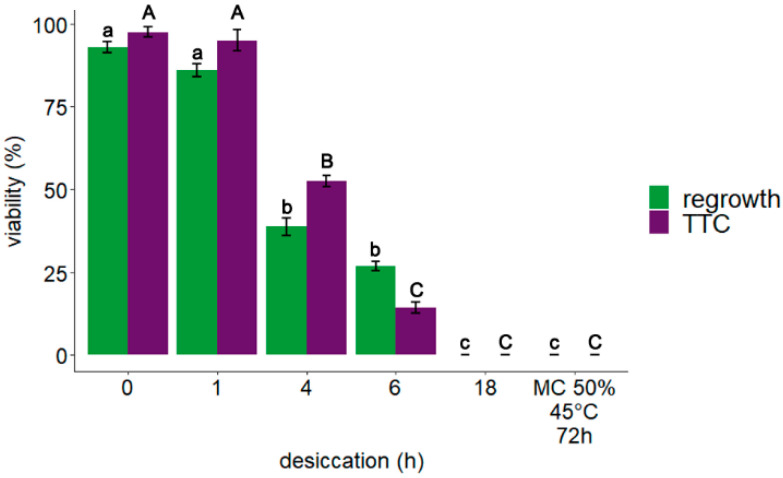
Effect of desiccation on *Acer pseudoplatanus* L. embryonic axes regrowth (*n* = 5) and respiratory activity measured by TTC assay (*n* = 4) [43]. The presented graph is supplemented with data obtained after storage of embryonic axes in accelerated aging conditions (72 h, 45 °C, MC of 50%). Statistical analyses were conducted on separate data sets. Capital and small letters correspond to analysis of TTC staining and tissue regrowth, respectively. Values labelled with different letters are significantly different at *p* ≤ 0.05, according to the Duncan test. Data represents mean ± SE.

**Figure 2 cells-11-01748-f002:**
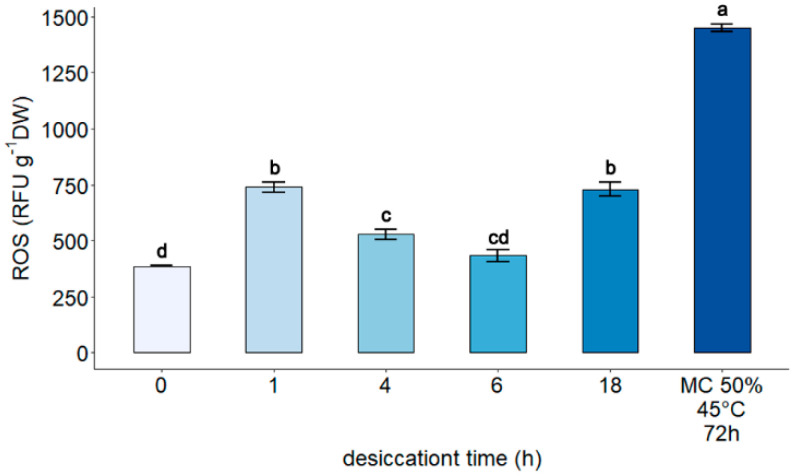
Influence of gradual desiccation on ROS level in embryonic axes of *Acer pseudoplatanus* L. [43]. The presented graph is supplemented with data obtained after storage of embryonic axes in accelerated aging conditions (72 h, 45 °C, MC of 50%). Values labelled with different letters are significantly different at *p* ≤ 0.05, according to the Duncan test. Data represents mean ± SE, *n* = 4.

**Figure 3 cells-11-01748-f003:**
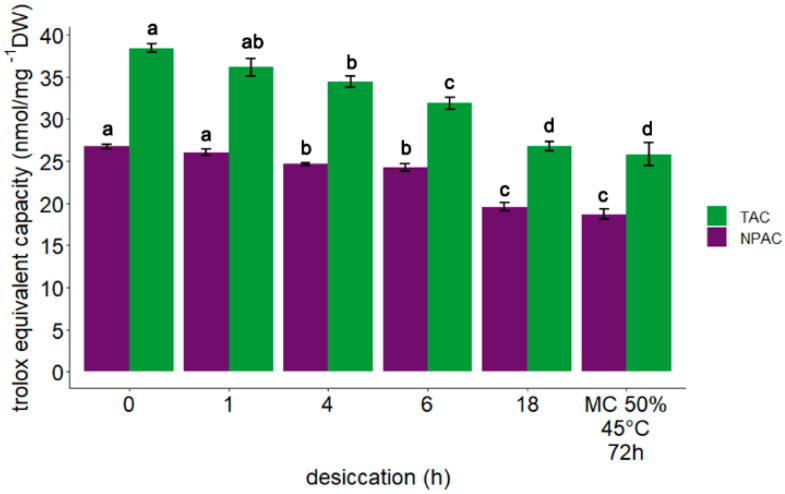
Influence of gradual desiccation and accelerated aging (72 h, 45 °C, MC of 50%) on total antioxidants capacity (TAC) and non-protein antioxidant capacity (NPAC) in embryonic axes of *Acer pseudoplatanus* L. Values labelled with different letters are significantly different at *p* ≤ 0.05, according to the Duncan test. Data represents mean ± SE, *n* = 4.

**Figure 4 cells-11-01748-f004:**
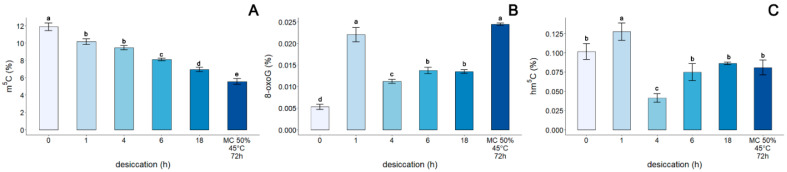
Influence of gradual desiccation and accelerated aging conditions (72 h, 45 °C, MC of 50%) on the relative global level of (**A**) 5-methylcytosine; (**B**) 8-oxo-7,8-dihydroguanine; (**C**) 5-hydroxymethycytosine. Values labelled with different letters are significantly different at *p* ≤ 0.05, according to the Duncan test. Data represents mean ± SE, *n* = 3.

**Figure 5 cells-11-01748-f005:**
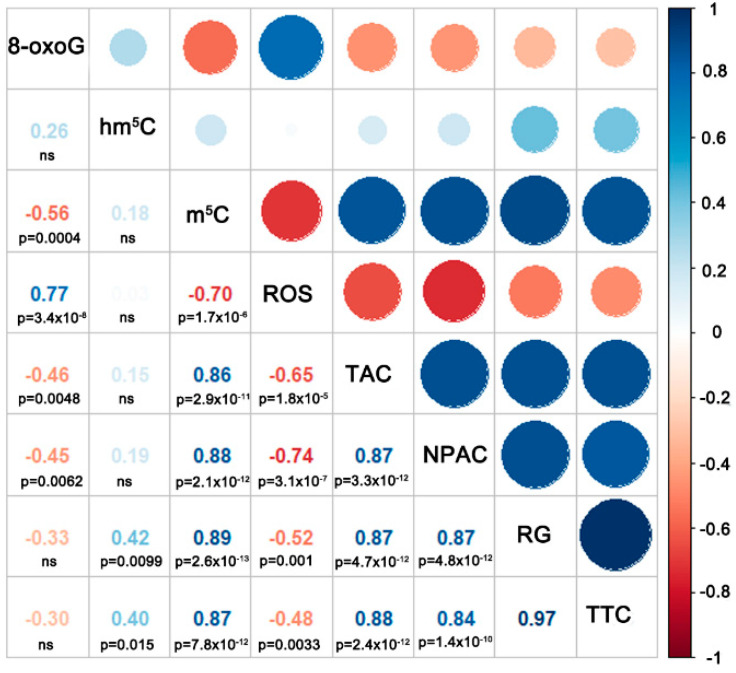
The Spearman correlation coefficient between the means of oxidative stress measurements (ROS), viability measurements (TTC, regrowth RG), global level of m^5^C, hm^5^C, 8-oxoG, total antioxidant capacity (TAC), non-protein antioxidant capacity (NPAC) measured in *Acer pseudoplatanus* L. embryonic axes directly after desiccation or accelerated aging (72 h, 45 °C, MC of 50%). The size of the circles represents the level of correlation (r), bigger circles indicate that a given trait correlates at higher level. Blue color indicates positive correlations and red indicates negative correlations.

**Figure 6 cells-11-01748-f006:**
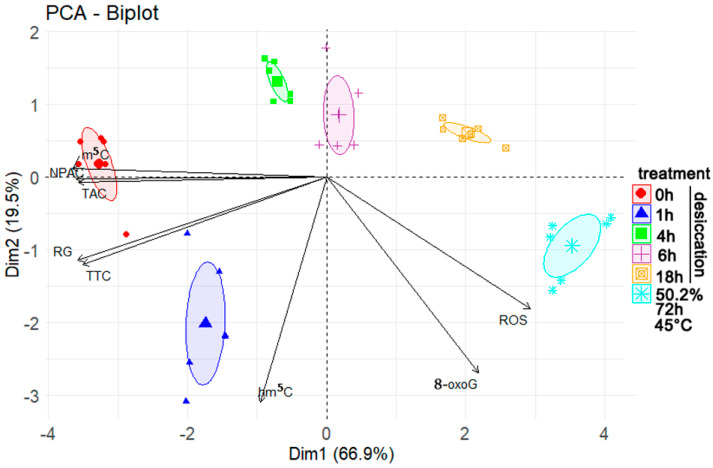
Results of principal component analysis (PCA) applied on the correlations of regrowth (RG), metabolic activity (TTC), ROS level, relative amount (%) of m^5^C, hm^5^C, 8-oxoG, total antioxidant capacity (TAC), non-protein antioxidant capacity (NPAC) in embryonic axes directly after desiccation or accelerated aging (72 h, 45 °C, MC of 50%). Ellipse confidence level = 0.95.

**Table 1 cells-11-01748-t001:** Moisture content (MC) and water content (WC) of *Acer pseudoplatanus* L. embryonic axes after desiccation between 1 and 18 h.

Desiccation (h)	MC, % (WC, g H_2_O g^−1^ Dry Weight)
0	50.2 ± 0.08
	(1.01 ± 0.03)
1	19.3 ± 0.07
	(0.24 ± 0.01)
4	11.9 ± 0.08
	(0.13 ± 0.01)
6	9.2 ± 0.4
	(0.10 ± 0.01)
18	5.7 ± 1.2
	(0.06 ± 0.01)

## Data Availability

The data presented in this study are available on request from the corresponding author.

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
