# Peer review of "Epigenetic Marks, DNA Damage Markers, or Both? The Impact of Desiccation and Accelerated Aging on Nucleobase Modifications in Plant Genomic DNA"

_cells, 2022, doi:10.3390/cells11111748_

Round 1
Reviewer 1 Report
This study investigated the possible correlation between the level of three DNA modifications (m5C, hm5C, and 8-oxoG) and the level of ROS and antioxidants (AOX) in embryonic axes of sycamore maple under desiccation stress conditions and after accelerated aging. Although it is interesting to show that tissue desiccation induces a similar pattern of changes in the global level of hm5C and 8-oxoG, the molecular mechanism underlining is unclear. I suggest that the authors can also analyze the gene expression and /or enzyme activities related to DNA modification and ROS/AOX changes in embryonic axes under desiccation stress conditions, which could provide crucial insight about how DNA modification and ROS/AOX changes are established.
Author Response
Comment: This study investigated the possible correlation between the level of three DNA modifications (m5C, hm5C, and 8-oxoG) and the level of ROS and antioxidants (AOX) in embryonic axes of sycamore maple under desiccation stress conditions and after accelerated aging. Although it is interesting to show that tissue desiccation induces a similar pattern of changes in the global level of hm5C and 8-oxoG, the molecular mechanism underlining is unclear. I suggest that the authors can also analyze the gene expression and /or enzyme activities related to DNA modification and ROS/AOX changes in embryonic axes under desiccation stress conditions, which could provide crucial insight about how DNA modification and ROS/AOX changes are established.
Our response: We appreciate the comment from the Reviewer and agree that the presented area of research is interesting and worth expanding with additional experiments and techniques in order to provide more information. However, we need to explain why some of her/his suggestions were not followed. First of all, there is no possibility of testing gene expression as no genome or transcriptome sequences are currently accessible for Acer pseudoplatanus. Therefore, it is not possible to design valid primers for RT-qPCR experiments. Secondly, as we were interested in general oxidative status of cells, thus antioxidant capacity was assayed. In that test both small-molecule antioxidant capacity and protein antioxidant capacity were assayed. Consequently, in this assay also activity of enzymatic antioxidants was monitored. Additionally, in modified version of the assay, only small molecule non-protein antioxidant capacity was analyzed separately. Therefore, the difference between non-protein antioxidant capacity and total antioxidant capacity represents antioxidant capacity of protein fraction what is visible on the Figure 3. Thanks to such approach the entire antioxidant status of desiccated cells was monitored and we believe it provided valid information about general redox milieu. Of course, tracking activity of separated enzymes would provide information how each of them is affected by desiccation nevertheless, we would not expect a significant difference in understanding of the overall cellular redox state at each desiccation step. However, the suggestion from Reviewer is interesting, and provides us direction for further, deepened research. Thirdly, all experiments were performed on seeds that were collected from strictly selected natural stands, what significantly limits the amount of accessible research material. Moreover, seeds older than 8 weeks after collection were not used in the experiments, as maple recalcitrant seeds age during storage, and prolonged storage would affect all measured parameters. We are also not able to conduct the suggested experiments currently, as seed of Acer pseudoplatanus mature in September/October. Therefore, when planning experiments with such plant material it is crucial to choose precisely the parameters that are desired to be tracked. We agree that experiments proposed by Reviewer would be highly informative, however due to circumstances mentioned above they were not possible to perform or their aim would be beyond planned experimental works. Our major goal was to track simultaneously three DNA modifications to firstly: analyze changes in their amount during desiccation and ageing, secondly: to indicate which one of them would be the modification of choice for seeds viability testing. We feel we successfully completed the research goals and we indicated that only 5-methylcytosine showed the potential to become a valid seed viability marker. However, we also stated that based on current results we expect hm5C to origin from ROS activity.
Therefore, we hope that our response clarifies our choices and experimental planning as well as we hope to get the Reviewer’s approval.
Reviewer 2 Report
The manuscript reports relevant results regarding the methylation level of C and the presence of oxidized nucleobases in DNA samples of embryo tissues subjected to desiccation and accelerated ageing. The work has been carried out with Acer pseudoplatanus embryonic axes, particularly sensitive to desiccation as this species has recalcitrant seeds. This report is especially important as the number of published works regarding methylation status and oxidative damage during seed conservation are still scarce.
The manuscript is very well written and only in very few occasions some language editing has been suggested (please see marks/comments in the reviewed manuscript).
I have also included some comments in the manuscript; some of them are the following (please check the revised manuscript to see all of them):
- Line 22: “We demonstrate that tissue desiccation induces a similar pattern of changes in the global level of hm5C and 8-oxoG”; I do not think this is correct as no significant correlation was observed between the level of hm5C and 8-oxoG
- Table 1: Unit for WC should be “g H2O g-1 dry weight”
- Lines 161-163: Details of in vitro procedure are missing. Were they similar to ref [43]?
If so, please state.
- Lines 217-223: It seems that both sentences refer to the same correlations carried out and shown in Fig. 5.
- Lines 235-237: It seems the data of the regrowth and TTC tests of the desiccated embryonic axes have already been published in [43]. So, this sentence is not clear as indicates “results from the present study ….”. It is not clear if the data of regrowth are exactly the same as in [43].
- Line 250: I guess you mean Fig. 5. As figures should be mentioned in order, you should write "(see Fig. 5)”, or you should not mention the correlations now, as there is a section regarding the correlations.
- Line 272: As figures should be mentioned in order, you should write "(see Fig. 5), or you should not mention the correlations now, as there is a section regarding the correlations.
- Line 351: Recalcitrant seeds are sensitive to dehydration and do not undergo much desiccation prior shedding. The expression "sufficient maturation-drying" is not clear.
- Lines 411-413: I believe that this sentence states a too general principle ("plant tissues" with reduction in viability show a decline in DNA methylation), that cannot be supported by the research reported or the two mentioned references. All these studies have been carried out with seeds or embryos. I think the sentence should be referred only to "embryo tissues".
- Lines 511-513: I think this sentence needs a reference.
Several times in the discussion, papers by the same group are mentioned (self-citations) supporting their new findings regarding studies with seeds/embryos. I would appreciate if authors could make an effort to cite other groups that have worked on methylation/oxidative stress in this type of plant material.

Author Response
We addressed all comments provided by Reviewer in the letter and directly in the manuscript’s commentary section. All of them can be followed in corrected manuscript and in point-by-point response below.
Comment: - Line 22: “We demonstrate that tissue desiccation induces a similar pattern of changes in the global level of hm5C and 8-oxoG”; I do not think this is correct as no significant correlation was observed between the level of hm5C and 8-oxoG
Our response: Our results regarding desiccation-driven global changes in the levels of 8-oxoG and hm5C are presented in figure 4. When we were tacking the changes step by step, it became clear for us that after 1 h of desiccation the increase in both modifications was detected. However, the relative change in 8-oxoG amount was higher than for hm5C what is in concordance with the lowest redox potential of G among all nucleobases. Next, a decline in the amount of both modifications were detected, which we explained as a result of ongoing base excision repair process [1]. In both cases, that decline was followed by increase observed in embryos of low and lack of viability. Therefore, we stated that the trend of changes in both modifications is similar, although some differences occur. We explained the differences between 8-oxoG and hm5C as a result of dynamic changes in m5C amount, that is a substrate for oxidation and it was detected to decrease during subsequent steps of desiccation. Moreover, hm5C can be further oxidized is plant tissues. Therefore, changes in substrate and further oxidation processes should be taken into account as they may influence the finale level of hm5C. On the other side, no change in G, as a substrate for oxidation, is expected. Consequently, after careful considerations we sustain our statement, that global levels of both modifications undergo similar changes when tissues are affected by desiccation causing an oxidative stress showed by ROS and antioxidant measurements.
It is important to clarify as well, that Spearman's correlation measures the strength and direction of monotonic association between two variables. Therefore, to show significant correlations, the same type of change as well as similar levels of changes must occur. In our case we did not observe such monotonic changes, consequently no Spearman's correlation was detected between hm5C and other measured parameters.
Nevertheless, we have modified the sentence as follows: “We demonstrate that tissue desiccation induces a similar trend in changes in the global level of hm5C and 8-oxoG, which may suggest that they both originate from the activity of reactive oxygen species (ROS).”
- Plitta-Michalak, B.P.; Ramos, A.A.; Pupel, P.; Michalak, M. Oxidative Damage and DNA Repair in Desiccated Recalcitrant Embryonic Axes of Acer pseudoplatanus L. BMC Plant Biol. 2022, 22, 40, doi:10.1186/s12870-021-03419-2.
Comment: - - Table 1: Unit for WC should be “g H2O g-1 dry weight”
Our response: Corrected.
Comment: - - Lines 161-163: Details of in vitro procedure are missing. Were they similar to ref [43]? If so, please state.
Our response: Corrected.
The sentence has been modified as follows: “An in vitro regrowth assay was conducted as previously described [43] using five biological replicates comprising 10-15 embryonic axes in each replicate.”
Comment: - - Lines 217-223: It seems that both sentences refer to the same correlations carried out and shown in Fig. 5.
Our response: Corrected.
Comment:- Lines 235-237: It seems the data of the regrowth and TTC tests of the desiccated embryonic axes have already been published in [43]. So, this sentence is not clear as indicates “results from the present study ….”. It is not clear if the data of regrowth are exactly the same as in [43].
Our response: Corrected.
The sentence has been modified as follows: “Data obtained from a previous study [43], along with data for metabolic activity of embryonic axes subjected to accelerated ageing, are also presented (Figure 1). Altogether results indicated that a desiccation period of four and six hours significantly decreased the viability of embryonic axes while explants desiccated for 18 h, or subjected to accelerated aging, were dead.”
Comment:- Line 250: I guess you mean Fig. 5. As figures should be mentioned in order, you should write "(see Fig. 5)”, or you should not mention the correlations now, as there is a section regarding the correlations.
Our response: Corrected.
Comment:- - Line 272: As figures should be mentioned in order, you should write "(see Fig. 5), or you should not mention the correlations now, as there is a section regarding the correlations.
Our response: Corrected.
Comment:- - Line 351: Recalcitrant seeds are sensitive to dehydration and do not undergo much desiccation prior shedding. The expression "sufficient maturation-drying" is not clear.
Our response: As this sentence could have been confusing and unclear, we have decided to remove it.
Comment:- - Lines 411-413: I believe that this sentence states a too general principle ("plant tissues" with reduction in viability show a decline in DNA methylation), that cannot be supported by the research reported or the two mentioned references. All these studies have been carried out with seeds or embryos. I think the sentence should be referred only to "embryo tissues".
Our response: We have agreed with the Reviewer. The sentence has been corrected as follows:
“Collectively our data indicate that the decline in DNA methylation that occurs in plant embryos that also undergo a reduction in viability is a universal process resulting from being subjected to different stresses (water withdrawal vs. high temperature and high MC).”
Comment:- - Lines 511-513: I think this sentence needs a reference.
Our response. We can not provide the reference, as this statement is proposed by us based on our comparative results on m5C and hm5C changes observed in embryos. Because we detected a decrease in the amount of m5C after accelerated aging therefore, the amount of substate for oxidation was lower. Despite our efforts, we have not found any other literature that would provide other explanation.
Comment: Several times in the discussion, papers by the same group are mentioned (self-citations) supporting their new findings regarding studies with seeds/embryos. I would appreciate if authors could make an effort to cite other groups that have worked on methylation/oxidative stress in this type of plant material.
Our response. We have added several citations to relevant literature. We would like to thank for giving us notice about this oversight.
Comment: Total antioxidant capacity (TAC) was measured using a commercially-available Total Antioxidant Capacity Assay kit.
Our response: The sentence has been corrected as follows: Total antioxidant capacity (TAC) was measured using a commercially-available Total Antioxidant Capacity Assay kit (Merck, Darmstadt, Germany).
We would like to thank the Reviewer for the contribution to perfecting the manuscript. We have improved the manuscript according to the suggestions. We hope to get the Reviewer’s approval.
Round 2
Reviewer 1 Report
The authors have adequately addressed my previous comments.